# Uncertainty handling in intra-operative multispectral imaging with invertible neural networks

**Tim J. Adler**[1,2]                                                    T.ADLER@DKFZ.DE

**Lynton Ardizzone**[3]**, Leonardo Ayala**[1]**, Janek Gröhl**[1,4]**, Anant Vemuri**[1]**, Sebastian J. Wirkert**[1]**, Beat P. Müller-Stich**[5]**, Carsten Rother**[3]**, Ullrich Köthe**[3]**,**

**Lena Maier-Hein**[1]                                                  L.MAIER-HEIN@DKFZ.DE

[1] *Division Computer Assisted Medical Interventions (CAMI), German Cancer Research Center (DKFZ), Heidelberg, DE*
[2] *Faculty of Mathematics and Computer Science, Heidelberg University, DE*
[3] *Visual Learning Lab, Heidelberg University, DE*
[4] *Medical Faculty, Heidelberg University, DE*
[5] *Division of Minimally-invasive Surgery of the Department of General Surgery, Heidelberg University, DE*

**Editors:** Under Review for MIDL 2019

**Keywords:** surgical data science, multispectral imaging, ambiguity, uncertainty estimation, deep learning, invertible neural networks, out of distribution detection

## 1. Introduction

Replacing traditional open surgery with minimally-invasive techniques for complicated interventions such as partial tumor resection or anastomosis is one of the most important challenges in modern healthcare. In these and many other procedures, characterization of the tissue remains challenging by means of visual inspection. Conventional laparoscopes are limited by imitating the human eye, recording three wide color bands (red, green and blue). In contrast, multispectral cameras remove this arbitrary restriction, and allow for the capture of many narrower bands of light. In previous work, we have shown that the inverse problem of converting pixel-wise multispectral measurements to underlying tissue properties can be addressed with machine learning techniques (Wirkert et al., 2016, 2017). Key remaining challenges are related to the handling of uncertainties: (1) Machine learning based approaches can only guarantee their performance on data that is sufficiently similar to the training data, hence, *out of distribution* (OoD) samples have to be handled and (2) most algorithms provide a point estimate of a physiological parameter (e.g. blood oxygenation $sO_2$), neglecting the fact that the problem may be inherently ambiguous (i.e. two radically different tissue parameter configurations may – in theory – result in similar measured spectra). We tackle these shortcomings with a framework for uncertainty handling (Figure 1) that involves (1) filtering OoD samples and then (2) computing the full posterior probability distribution for tissue parameters given the measured spectra. Analysis of the posteriors not only provides us with a means for quantifying the uncertainty related to a

specific measurement but also enables a fundamental theoretical analysis about which tissue properties can in principle be recovered with multispectral imaging (MSI).

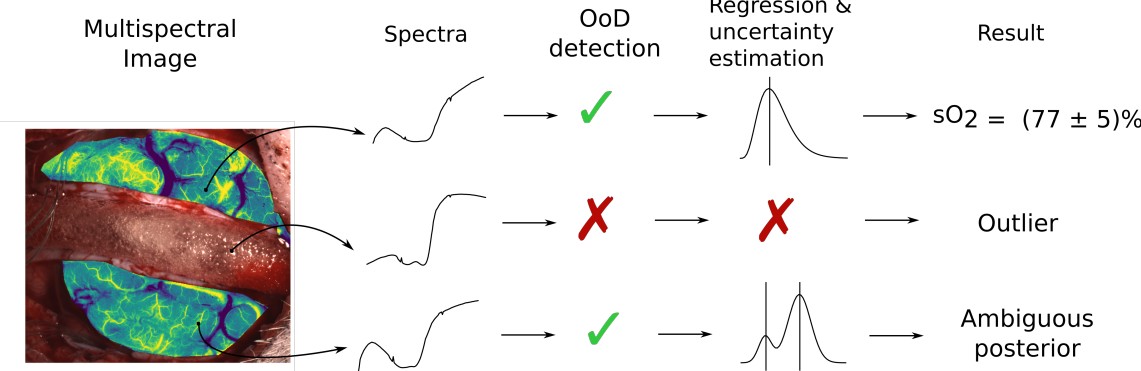

Figure 1: Proposed multi-stage process for uncertainty handling in multispectral image analysis.

## 2. Methods

Our proposed approach to physiological parameter estimation from MSI measurements is based on generation of training data using a Monte Carlo based simulation framework (Wirkert et al., 2016) and training an *invertible neural network* (INN) to convert MSI measurements to underlying tissue properties. During test time, the pixel-wise MSI measurements are first filtered based on an OoD check and then converted to full posteriors representing the probability of a physiological tissue parameter(-set) given the measurement. The core components are the following:

**Invertible Neural Network:** The core of our method are INNs, which were presented in detail in previous work (Dinh et al., 2016) and applied to inverse problems in (Ardizzone et al., 2019). These networks have the advantage that they provide a full posterior distribution for the estimated parameter rather than a point estimate, thus allowing us to check if the regression is unique (i. e. has a sharp peak) or ambiguous (i. e. the posterior has multiple modes).

**Out of distribution detection:** Machine learning algorithms are typically designed to compute a result regardless of the input data. This can lead to spurious output as the accuracy of the algorithms can only be guaranteed on input sufficiently similar to the training data. Hence, it is necessary detect OoD samples. To tackle this problem, we build upon the work of Choi et al. (2018) who proposed to use the *widely applicable information criterion* (WAIC) (Watanabe, 2009) as a metric for outliers. In order to compute this quantity, we need an ensemble of parametric models approximating the log-probability distribution of the training data. As INNs have tractable Jacobians they fulfill this requirement. Our framework thus involves an ensemble of INNs whose outputs are used to estimate the WAIC.

## 3. Experiments, Results and Discussion

**Ambiguity of regression task:** We used the INN concept for a fundamental analysis on which tissue properties can be recovered without ambiguity depending on the MSI camera design. As detailed in Adler et al. (2019), we compared multiple cameras in medical use with regard to the well-posedness of the task of estimating oxygenation $sO_2$ and blood volume fraction $v_{HB}$. According to our experiments, the uncertainty rises as the number of bands decreases, and our method is able to detect this. Figure 2a illustrates that the posterior generated by a 3-band camera is wide compared to the 8-band posterior and even multi-modal leading to a worse predictive performance.

**Out of distribution detection:** We used simulated spectra (Wirkert et al., 2016, 2017) to train an ensemble of five INNs for OoD detection. The simulation framework incorporates a tissue model assuming blood and water as the main absorbers. We compared the WAIC of the simulated spectra to porcine organ spectra which we classified manually as *in domain* (iD) or *out of domain* (oD) depending on the tissue containing absorbers not considered in our simulation framework (e. g. bilirubin in gallbladder). Our hypothesis was that the WAIC metric should be able to separate oD from iD organs. The experiment demonstrated the following two points: Firstly, we found that the mean WAIC values of the iD organs were substantially higher than those of the simulated test data, indicating a remaining domain gap. Secondly, the WAIC values of the oD organs were the highest overall, far above the iD values. This confirms the hypothesis and allows us to use the WAIC to discriminate between in and out of distribution data. A comparison of the test, iD and oD data can be found in Figure 2b.

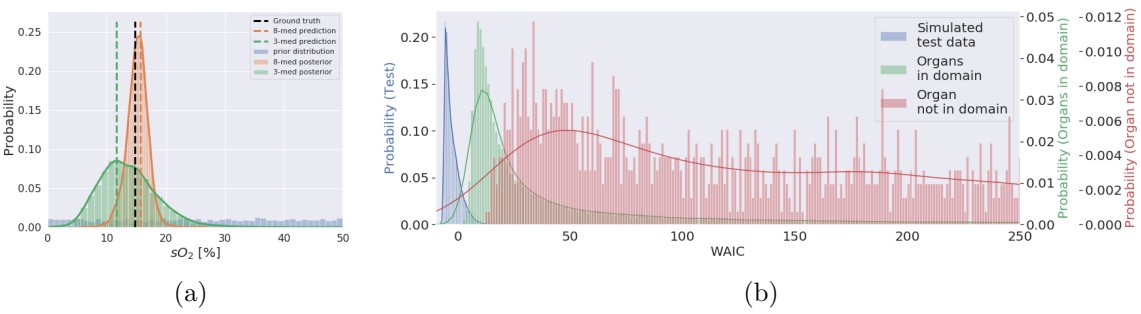

(a)                                    (b)

Figure 2: (a) Example of two posterior distributions as provided by our INN (source: (Adler et al., 2019)). (b) Histograms of WAICS for simulated test data as well as *in vivo* in distribution and out ouf distribution data (tails truncated).

## 4. Conclusion

In conclusion, this work presents a novel framework for intra-operative multispectral imaging taking into account both uncertainty of the data distribution (OoD detection) and uncertainty related to the well-posedness of the model (ambiguity in regression task). We think that our approach can lead to more robust estimators for morphological and functional imaging tasks, bringing the field of MSI one step closer from the bench to the bed side.

## Acknowledgments

We acknowledge support from the European Union through the ERC starting grant COM-BIOSCOPY under the New Horizon Framework Programme under grant agreement ERC-2015-StG-37960.

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
