# OpenReview forum: "Uncertainty handling in intra-operative multispectral imaging with invertible neural networks"
_MIDL.io/2019/Conference/Abstract — MIDL Abstract 2019_

### Official Review · AnonReviewer1 · 2019-05-01
**Unclear but interesting**

**Rating:** 3
**Confidence:** 3

**Review:**

The motivation of the application seems unclear and should be written more directly.

The abstract discusses invertible neural networks and their use in out of distribution detection.  It is unclear if the input is an image patch or a distribution of frequencies (I lean to distribution of frequencies given the figure but it should be more clear. What is the dimension?). What is the likelihood computed on? It should be clearly stated. The motivation for why the work uses an INN is does not include evidence for why this type of model is required and how other methods failed.

In any case the idea is interesting and the work could contain insights into difficult deployments using many different deep learning tools.

---

### Official Review · AnonReviewer2 · 2019-05-06

**Rating:** 4
**Confidence:** 3

**Review:**

The authors propose using invertible neural network (INN) to estimating oxygenation in tissue, using INNs allows the authors to compute posterior distributions as well as to calculate the ``widely applicable information criterion'' which they use for outlier detection. The authors show that 8-band cameras produce a unimodal distribution peaked near the ground truth, while 3-band cameras produce a bimodal posterior, where MAP estimation picks the incorrect mode, an obvious argument for why posterior distributions are important and of interest to the field.

---

### Decision · Program_Chairs · 2019-05-06
**Acceptance Decision**

Accept